# Perceived speech decoding and neurophysio-logical knowledge mining with explainable AI and non-invasive brain activity recordings

## Abstract

Explainable artificial intelligence (XAI) is a branch of AI directed at the development of machine learning (ML) solutions that can be comprehended by the human users. Here we use an interpretable and domain-grounded machine learning architecture applied to non-invasive magnetoencephalographic (MEG) data of subjects performing a speech listening task and discover neurophsyologically plausible spatial-temporal neuronal representations of latent sources identified through self-supervised network training process. Achieving high decoding accuracy in the downstream task our solution bridges the gap between high performance and big data-based AI and the classical neuroimaging research and represents a novel knowledge mining platform where the decoding rule can be interpreted using the accepted in electrophysiology terms and concepts which is likely to advance neuroscientific research.

## 1 Introduction

Explainable artificial intelligence (XAI) is a branch of AI focused on the development of the white-box machine learning (ML) solutions that can be comprehended by the human users. White-box models are easier to deploy in real-life tasks. They also enable knowledge discovery by rendering mechanistic explanations of the learned decision rules using domain-specific concepts and terminology when employed in a research setting.

Advancing towards the integration of XAI in neuroscience, Petrosyan et al. (2021) developed an interpretable machine learning framework and applied it to invasive recordings of brain activity during the speech production in (Petrosyan et al., 2022). By applying the proposed estimation theory based approach to the spatial and temporal convolution filter weights they managed to successfully reveal the localization of language function which spatially matched cortical sites whose subsequent stimulation led to speech arrest and speech production errors in both patients. Importantly, the neuronal activity's spectral signatures obtained through the proposed interpretation algorithm appeared physiologically plausible (Miller et al., 2007). This study can be considered as a showcase of using XAI for knowledge mining when a decoder trained to do a specific downstream task (speech decoding) allows for a mechanistic explanation, in this case using the electrophysiological terms of neural source location and the dynamic properties of its electrical activity. At the same time, invasive recordings provide high-quality data but are limited because of the sampling of brain activity at a small subset of locations. This precludes investigation into the macro-scale cortical mechanisms underlying the explored function.

Magnetoencephalography (MEG) (Cohen, 1968), a contactless whole brain functional imaging modality offers unique opportunities for tracing neuronal processes with millisecond-scale temporal resolution. Recently, a universal deep learning (DL) model was built to successfully decode 3 second long audited speech segments (out of more than 1,000 possibilities) based on the MEG signals registered during the speech perception in multiple volunteers (Défossez et al., 2023). The demonstrated decoding of the perceived speech is a significantly simpler task than that of predicting speech production from brain activity recordings. However, since it is now accepted in the community that language comprehension and speech production share similar neural circuits (Fadiga et al., 2002; Liu et al., 2023; Eliades & Wang, 2008), the results obtained in the passive auditory paradigms may

shed light on the cortical mechanisms of speech representation and production and benefit not only the fundamental neurolinguistic research but also the development of long awaited speech prosthetic devices.

To this end Défossez et al. (2023) provided some interpretation of the decoder weights and showed that dominant attention was paid to the lateral MEG sensors proximal to the primary auditory cortices. At the same time, due to inherently non-linear relationship, the weights of the spatial attention layer can not be rigorously converted to the underlying cortical sources to fully benefit from the MEG's high spatial resolving power. Endowed with the appropriate inverse operator (Bonaiuto et al., 2018; Wens, 2023), non-invasively collected multivariate MEG signals can be mapped back to the cortex to resolve neuronal sources located at a sub-centimeter distance.

The proposed network's subject block contains the subject adaptation layer which uses linear spatial processing to efficiently enable the aggregation of data from multiple subjects. However, the subject block lacks a temporal filtering layer which limits the subsequent interpretation of the decision rule. The use of temporal convolutions allows the network to benefit from the unprecedented temporal resolving power of MEG. Analyzing temporal correlation weights we can unravel the brain's rhythmic activity hierarchy bearing the information within the context of the decoding task.

Finally, the proposed architecture utilizes a physically unjustified receptive field model created on the basis of a set of 2D Fourier harmonics which may adversely affect the compactness of the proposed solution.

To ameliorate the described shortcomings of the otherwise breakthrough solution developed by Défossez et al. (2023) and to illustrate the use of XAI methodology we modify the original architecture to enable the interpretability of the decision rule learned by the network. We also replace the heuristic spatial attention layer with that based on the spherical harmonics as the natural basis for capturing the geometry of the magnetic field as captured by the 3D sensor array and experiment with the interpretable subject block dimension to minimize the number of trainable parameters. Finally we add a temporal filtering layer to benefit from the temporal resolving power of MEG.

Using one of the two MEG datasets employed in (Défossez et al., 2023), the dataset described by Gwilliams et al. (2023), we show that the modified and simplified architecture performs comparably to the original network and at the same time allows for identifying neuronal sources pivotal for the decoding task along with dynamical properties of their activity. Reduction of the number of spatial channels from 270 to 6 in combination with the 3D spatial layer does not reduce but even slightly boosts the decoding accuracy.

We interpret the network's weights into the cortical distributions derived from the spatial patterns. We also derive spectral profiles of the electrical activity of latent sources to show that the obtained classification performance is supported by the rhythmic activity of sources in the primary auditory cortices, parietal (Wernicke) and frontal (Broca) cortical areas. Joint analysis of the power spectral density profiles and the associated spatial patterns reveals the dominant role of oscillatory activity in the alpha and beta bands. Intriguingly, our analysis shows that in addition to the sources of neuronal origin the network's interpretable subject block tunes of eye-movements as a latent source informative for the downstream decoding task. This is an exciting demonstration how XAI trained within the auditory downstream task reveals the well known link between mutually cooperative performance of visual and audio perception systems (Mendelson et al., 1976) related to tracking mentally constructed sentences during speech listening in the absence of any sentence-related visual or prosodic cues (Jin et al., 2018).

## 2 THEORETICAL BACKGROUND

### 2.1 GENERATING EQUATION MEG

The primary sources of MEG signals are the electric currents floating in the dendrites of pyramidal neurons that receive the excitation from the neuronal populations situated nearby in the other cortical layers. The bundle of mutually parallel dendrites of a large number of neurons occupying the cortical area of several tens of square millimeters is then approximated by a single *equivalent current dipole* (ECD) with location vector $\boldsymbol{r}_n = [x_n, y_n, z_n]^\top$ and orientation $\boldsymbol{\theta}_n = [\theta_n, \theta_n, \theta_n]^\top$ where $n$ denotes

the index of a neuronal source. Time varying dipole moment of the $n$-th ECD is called activation time series and is denoted as $s_n(t)$.

The array of $M$ MEG sensors surrounding the head and located at a set of locations $\boldsymbol{r}_m$, $m = 1, ..., M$, at each time instance $t$ measures a vector $\mathbf{x}(t) = [x_1(t), ..., x_M(t)]^\top$ and spatially samples the weak magnetic field produced by the superposition of magnetic fields generated by the vector $\mathbf{s}(t)$ of ECD activation moments $\mathbf{s}(t) = [s_1(t), ..., s_N(t)]^\top$, $n = 1, ..., N$, where $N$ is the number of active neuronal sources.

The MEG (and EEG) signal vector generated by a single $n$-th neuronal source (an ECD) can be modeled simply as $\mathbf{x}(t) = \boldsymbol{g}_n s_n(t)$, where $\boldsymbol{g}_n = \boldsymbol{g}(\boldsymbol{r}_n, \boldsymbol{\theta}_n)$ is the $M \times 1$ gain vector mapping to $M$ MEG sensors the activity of the $n$-th unit dipole with orientation $\boldsymbol{\theta}_n$ located at $\boldsymbol{r}_n$. Vector $\boldsymbol{g}_n$ can be visualized by interpolating its values between sensor locations. Historically, the visualization used level lines, similar to those employed in topographic maps, and hence vector $\boldsymbol{g}_n$ is often called *topography* of a source at location $\boldsymbol{r}_n$ and orientation $\boldsymbol{\theta}_n$. Vectors $\boldsymbol{g}_n$, $n = [1, ..., N]$ for the grid of $N$ cortical locations (and orientations) are obtained by solving Maxwell equations for the head as a volume conductor on the basis of geometric information about location and orientation of MEG sensors and cortical sources. The former results from the head and MEG sensor array coregistration procedure and the latter is dictated by the nodes of a triangulated cortical surface mesh extracted from the volunteer's head MRI volume using readily available tools such as FreeSurfer (Fischl, 2012).

When multiple sources are active due to linearity of Maxwell's equations the MEG's generative model is written as a superposition of the contribution of each neuronal source as

$$\mathbf{x}(t) = [x_1(t), ..., x_M(t)]^\top = \sum_{n=1}^{N} \boldsymbol{g}_n s_n(t) + \mathbf{e}(t), \tag{1}$$

where $\mathbf{e}(t)$ is the observation noise vector accounting for the forward modeling errors and the sensor noise.

## 2.2 FROM MEG SENSOR SIGNALS TO SOURCE ACTIVITY

The ultimate goal of MEG as a functional neuroimaging modality is to gain access to the activity of neuronal sources $\mathbf{s}_k(t)$, $k = 1, ..., N$. Typically this is accomplished using a spatial filter $\boldsymbol{w}^\top$ tuned to recover a source with specific properties and suppress the contributions of the activity of the other sources simply by computing a weighted linear combination of channel time series $\mathbf{x}(t)$, i.e.

$$\hat{s}_k(t) = \boldsymbol{w}^\top \mathbf{x}(t) = \sum_{n=1}^{N} \boldsymbol{w}_k^\top \boldsymbol{g}_n s_n(t) + \boldsymbol{w}_k^\top \mathbf{e}(t), \tag{2}$$

The most straightforward property of a neuronal source is its geometric location and orientation given by the pair of vectors $(\boldsymbol{r}_k, \boldsymbol{\theta}_k)$. In this case the spatial filter $\boldsymbol{w}_k$ tuned to $(\boldsymbol{r}_k, \boldsymbol{\theta}_k)$ can be found as follows. We first use forward modeling to compute the associated spatial pattern (or topography) $\boldsymbol{g}_k = \boldsymbol{g}(\boldsymbol{r}_k, \boldsymbol{\theta}_k)$ of this source and then find the spatial filter using, for example, the linear minimum variance beamformer (LCMV) principle as $\boldsymbol{w}_k = (\boldsymbol{g}_k^\top \boldsymbol{R} \boldsymbol{g}_k)^{-1} \boldsymbol{R}^{-1} \boldsymbol{g}_k$ and since $\boldsymbol{g}_k^\top \boldsymbol{R} \boldsymbol{g}_k$ is a scalar we can deduce that $\boldsymbol{w} \sim \boldsymbol{R}^{-1} \boldsymbol{g}_k$. From this we can see that in a general case the spatial filter $\boldsymbol{w}_k$ is not collinear to the corresponding spatial pattern (topography) $\boldsymbol{g}_k$ of a source. This happens because the optimal signal-to-noise ratio in the estimate $\hat{s}_k(t)$ is achieved by not only tuning *to* the target source but also by tuning *away* from the interfering sources (to minimize the output variance) whose spatial distribution is encoded in the data covariance $\boldsymbol{R} = E\{\mathbf{x}(t)\mathbf{x}^\top(t)\}$.

Millisecond-scale temporal resolution of EEG and MEG makes these modalities unique in studying fast paced neuronal processes and neural oscillations in particular. Several spatial decomposition methods are designed to represent the multichannel EEG and MEG data as a superposition of rhythmic components with specific properties. For example, the technique called spatial spectral decomposition (SSD, (Nikulin et al., 2011)) finds several spatial filters $\boldsymbol{w}_k$, $k = 1, ..., K$ tuned to $K$ sources of activity with the highest rhythmic signal-to-noise ratio. Another example is the Source Power Co-modulation (SPoC) method described in Dähne et al. (2014) designed to find rhythmic

components whose power is correlated with an external behavioral variable e.g. volume of the perceived sound, screen intensity or the muscle activity strength. Note that unlike in the previous example where we started with topography $\boldsymbol{g}_k$ and then derived the corresponding spatial filter, here we solve an optimization problem and first arrive at the spatial filter $\boldsymbol{w}_k$ and then seek the topography vectors $\boldsymbol{g}_k$ corresponding to the discovered spatial filters. It is the topography vectors $\boldsymbol{g}_k$ (and not the filter weights vectors $\boldsymbol{w}_k$) that can be scrutinized with an inverse modeling procedure to locate the resultant source on the cortex (Kay, 1993; Haufe et al., 2014).

As shown in (Haufe et al., 2014) and provided that a spatial decomposition method yields the Wiener-optimal solution, the topography vector corresponding to a given spatial filter can be found as

$$\hat{\boldsymbol{g}}_k = \frac{1}{\sigma^2}\boldsymbol{R}\boldsymbol{w}_k \sim \boldsymbol{R}\boldsymbol{w}_k. \tag{3}$$

Since source variance $\sigma^2$ is not known in practice we typically rely on the normalized version of $\hat{\boldsymbol{g}}_k$.

Once the topography vectors are found we can then map them to the cortex using an inverse solver. The most straightforward way is to use the minimum norm (MNE) inverse operator $\boldsymbol{W}_{MNE}$ computed as

$$\boldsymbol{W}_{MNE} = \boldsymbol{G}^\top \left(\boldsymbol{G}\boldsymbol{G}^\top + \lambda\boldsymbol{I}\right)^{-1} \tag{4}$$

where $\boldsymbol{G} = [\boldsymbol{g}_1, \boldsymbol{g}_2, .., \boldsymbol{g}_N]$ is the $[M \times N]$ forward model matrix comprising $N$ topographies of the equivalent current dipole sources located in the nodes of the cortical mesh.

The cortical distribution of sources corresponding to the $k$-th component can then be found as

$$\hat{\boldsymbol{s}}_k = \boldsymbol{W}_{MNE}\hat{\boldsymbol{g}}_k \sim \boldsymbol{W}_{MNE}\boldsymbol{R}\hat{\boldsymbol{w}}_k \tag{5}$$

and visualized on the cortex by color-coding the absolute values of the elements of $\hat{\boldsymbol{s}}_k$.

Spatial decompositions are a powerful tool in the analysis of multichannel electrophysiological data. However, the transformation they are capable of learning is limited to a mere linear combination of the data channels. More recently Petrosyan et al. (2021) introduced an interpretable deep neural network whose initial layers comprises factorized spatial and temporal filters. The authors showed how the spatial and temporal weights of the network's first layers can be interpreted to reveal source topographies and power spectral density (PSD) profiles of the latent sources pivotal to solving the downstream task this network is trained on. The authors have also demonstrated how the initial layers of this architecture can be used as a part of a more complex network and yet successfully recover the pivotal cortical sites is demonstrated in phenomenological experiments with speech cortex mapping in patients with intractable epilepsy (Petrosyan et al., 2022).

The interpretable subject block of the proposed interpretable network can be considered as consisting of branches, each having its own spatial and temporal filter, see Figure 1. Each $k - th$ interpretable subject block branch with output $r_k(t)$ performs the following operation:

$$r_k(t) = f\left(\boldsymbol{w}_k^T\mathbf{x}(t) * \boldsymbol{h}_k(\tau)\right), \tag{6}$$

where $*$ denotes temporal 1D convolution operation and $\boldsymbol{h}_k(\tau)$ is the pulse response of the temporal convolution filter of the $k$-th branch.

In this case, the spatial and temporal processing is performed within the mutual context and therefore the strategy for forming spatial patterns, or topographies, from the spatial filter weights is modified as compared to (3):

$$\hat{\boldsymbol{g}}_k \sim \boldsymbol{R}_{h_k}\boldsymbol{w}_k \tag{7}$$

where $\boldsymbol{R}_{h_k} = E\{(\mathbf{x}(t) . * h_k(\tau))(\mathbf{x}(t) . * h_k(\tau))^\top\}$ - is the covariance of the multichannel data filtered with the temporal filter of the $k$-th branch $h_k(\tau)$ and $.*$ denotes temporal convolution applied to each channel separately. The intuition behind this expression is that the spatial filter while tuning

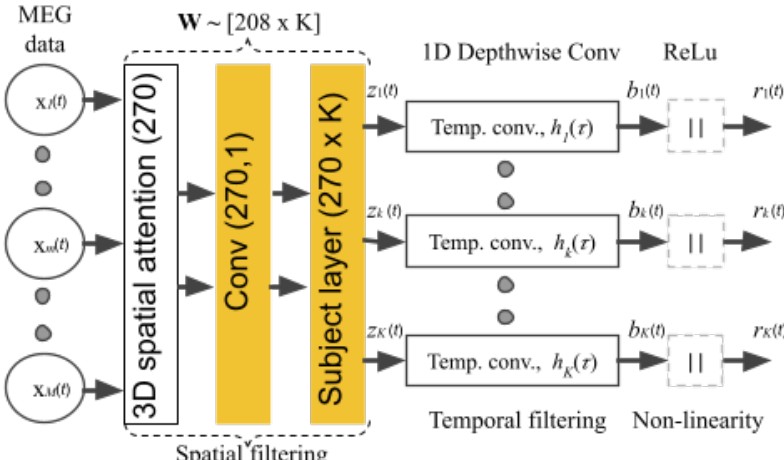

Figure 1: Interpretable subject block. Intact layers are denoted with sandy color as in the extended data Fig. 4.E of Défossez et al. (2023)

.

away from the interfering sources takes into account only those sources whose contribution was not eliminated by the corresponding temporal filter $h_k(\tau)$.

In much the same way we can obtain the power spectral density profiles $P_k(f)$ of the latent source the $k$-th branch is tuned to:

$$P_k(f) = H_k(f)Z_{\boldsymbol{w}_k}(f), \tag{8}$$

where $H_k(f) = FFT\{\boldsymbol{h}_k(\tau)\}$ is the frequency domain profile of the temporal filter of the $i$-th branch and $Z_{\boldsymbol{w}_k}(f) = FFT\{\boldsymbol{w}_k^\top \mathbf{x}(t)\} FFT^*\{\boldsymbol{w}_k^\top \mathbf{x}(t)\}$ is the power spectral density of the scalar time series $z_k(t) = \boldsymbol{w}_k^\top \mathbf{x}(t)$ obtained from the multichannel sensor time series data $\mathbf{x}(t)$ by the spatial filtering with branch-specific $\boldsymbol{w}_k$ and $^*$ denotes complex conjugation. Note that if we were to represent the temporal convolution as a matrix-vector product using the state-space approach based on the lagged temporal embedding, our expression (8) would more closely resemble that for the spatial pattern vs. filter relationship (7). In this case however the matrix and vector dimensions would correspond to the number of time lags of the state-space representation.

## 3 OUR SOLUTION DETAILS

### 3.1 3D SPATIAL ATTENTION LAYER

In their paper, Défossez et al. (2023) used spatial attention layer with parameterization in the Fourier space over 2D-projected sensor layout. However, since originally the sensors are located in 3D and form approximately a spherical shape, we propose parameterization with a set of 3D spherical harmonics (Sivakumar et al., 2016) as a more natural way to capture information about the geometric properties of the sensor array. To this end the receptive fields of our $J = 270$ secondary channels are formed by the vectors $\boldsymbol{a}_j^\top = [a_{j1}, .., a_{jm}, ..., a_{jM}]$, $m = 1, ..., M, j = 1, ..., J$ with elements parameterized through spherical harmonics as

$$a_{jm} = \sum_{l=1}^{L} \sum_{k=-l}^{l} z_j^{k,l} Y_l^k(\varphi_m, \theta_m), \tag{9}$$

where $Y_l^k(\phi_m, \theta_m)$ are the spherical harmonics evaluated on the unit sphere at the polar and longitude angles corresponding to the spherical coordinates of the $m$-th sensor, while $z_j^{k,l}$ are the learnable

parameters. Not only this approach allows us to take into account the actual non distorted sensor positions it also physically plausible since the magnetic field vectors generated by a spherical or close to spherical volume conductor can be compactly decomposed into a superposition of the gradients of distance-weighted spherical harmonics.

The spatial attention layer weights $\tilde{\boldsymbol{a}}$ are then computed as normalized softmax() transformed vectors $\tilde{\boldsymbol{a}}_j = softmax(\boldsymbol{a}_j), j = 1, .., J$ and the output of our 3D attention layer is then computed as

$$\text{SA}_j\left(\mathbf{x}(t)\right) = \tilde{\boldsymbol{a}}_j^\top \mathbf{x}(t), \ j = 1, .., J. \tag{10}$$

As well as in (Défossez et al., 2023) this layer performs linear processing of the input data vector $\mathbf{x}(t)$. Note that the authors of the original paper called this layer *Spatial attention* which we believe may be confusing especially in the context of the attention mechanism used by transformers where the weights depend on the input data and the overall processing becomes non-linear. Should this layer perform the classical attention operation this network would lose the straightforward interpretability exercised in our study.

## 3.2 TEMPORAL FILTERING

Brain rhythms are key components of non-invasively measured neuronal activity, reflecting different aspects of how the cortex processes incoming information (Buzsaki, 2006). By targeting specific frequency ranges using learnable temporal convolutions, decoding performance can be improved, while also enhancing the interpretability of the resulting decision rules. As shown in Figure 1 , we augmented the interpretable subject block of the network with temporal filters. To this end we used trainable 1D convolution filters of length 750 ms which corresponds to 75 taps given the sampling frequency $f_s = 100$ Hz that we downsampled the data to. Note that in the original paper the authors downsampled the data to 120 Hz.

The temporal convolution with trainable sequence $h_k(\tau)$ is performed within the $k$-th branch, $k = 1, .., K$ branches and is preceded by the spatial filter with coefficients taken as the $k$-th row of the aggregate spatial filter matrix $\boldsymbol{W}$. We refer to $h_k(\tau)$ as the pulse response of the temporal filter for the $k$-th branch. Unlike the subject layer, temporal filters were not subjects specific and were trained on all 27 subjects Note that given enough training data the added temporal filters should not adversely affect the decoding accuracy as in the case when the frequency band specificity is not required the network can learn $h_k(\tau) = \delta(\tau)$ to be all-pass filters implementing the identity transform. The temporal filter output is optionally processed with non-linearity shown with dashed squares in Figure 1.

Taken together the modified interpretable subject block of the network performs operation (6). Parameter $K$ corresponds to the number of branches in our interpretable subject block. Hypothetically, during the training process each branch gets tuned to a particular neuronal source active in the specific frequency range and characterized by a well defined spatial pattern (topography).

Our experiments (not described here) showed that the non-linearity caused a significant drop in performance. Therefore unlike Petrosyan et al. (2021) we do not use a non-linearity $f()$ past the temporal filter in the experiments reported here.

## 3.3 WEIGHTS INTERPRETATION

The spatial filter pertaining to each branch is calculated as the row of $[208 \times K]$ matrix $\boldsymbol{W}$ obtained by multiplying the matrices of coefficients of the *3D spatial attention layer* and the two subsequent layers left unmodified from the original solution except for the parameter $K$ controlling the number of branches in the interpretable subject block. Temporal filter weights are simply the coefficients of the temporal convolution layers $h_k(\tau), k = 1, .., K$. Using expressions (5) and based on the MNE inverse operator (4) computed using the forward model calculated based on the averaged cortical mesh. The neuronal sources generating MEG and EEG are the apical dendrites of the pyramidal neurons and are anatomically oriented perpendicularly to the cortical surface. Recovery of their orientation requires very dense sampling of the cortical mesh when it is extracted from a subject's MRI. This is often impractical especially when dealing with low quality MRI data and limited computational resources.

To gain robustness against the probable errors in the dipole orientation for each $n$-th location we included in the forward model $\boldsymbol{G}$ a set of three topography vectors corresponding to the three orthogonal orientations $\boldsymbol{G} = [\boldsymbol{g}_1^x, \boldsymbol{g}_1^y, \boldsymbol{g}_1^z, .., \boldsymbol{g}_N^x, \boldsymbol{g}_N^y, \boldsymbol{g}_N^z]$ of an ECD and computed the inverse operator $\boldsymbol{W}_{MNE}$ according to (4) with $\lambda = 0.1$.

Correspondingly, the inverse solution vector (5) has length $3N$ as it contains three elements per each $n$-th cortical location $\boldsymbol{s}_n = [s_n^x, s_n^y, s_n^z]^\top$. To visualize this vector for each $n$-th cortical site we computed L2-norm of the 3-dimensional vector $\rho_n = ||\boldsymbol{s}_n||$ pertaining to each cortical location and color-coded $||\boldsymbol{s}_n||$, $n = 1, .., N$ on the cortical mantle.

### 3.4 Training and testing

In this work we have first implemented our own version of the network described by Défossez et al. (2023) and applied it to one of the two MEG datasets described in the study by Gwilliams et al. (2023). This dataset contains audio and MEG signals recorded during the two identical 1 hour 30 minutes long sessions when 27 English-speaking participants listened to four fictional stories from the Masc corpus. The study was approved by the institutional review board ethics committee of New York University Abu Dhabi. We have used the entire three stories *lw1*, *cable spool fort*, *easy money* and also the first 5 pieces of *the black willow* story for training. For testing we used the last 7 pieces of *the black willow* story. This resulted in 2685 training and 999 testing segments which makes this study one of the few that utilize large datasets. Most of the previous works, e.g. Petrosyan et al. (2021; 2022), use smaller datasets. Also in our experiments we did not align the testing segments against word onset moments which represents potentially a more challenging setting than that reported in the original paper by Défossez et al. (2023) where the test set contained data segments synchronized to the word onset.

## 4 Results

### 4.1 Decoding accuracy

We assessed the model's efficacy by testing it on recordings associated with chapters 5 to 11 of "The Black Willow" story. One can see our results and ablations in Table 1. Firstly, we trained and evaluated the model presented by Défossez et al. (2023) to use it as a baseline. Note, all experiments besides this original baseline used 24 spatial harmonics in the attention layer. The baseline model used 32 harmonics as in the original paper. Then, to highlight the efficiency of our 3D spatial attention layer over the original 2D spatial attention we trained the same model only changing the spatial attention layer. It brought down the amount of parameters by almost 400K and allowed to improve the top-1 accuracy by 2.26 percent. Then, we trained our network with different variations of layers. All of them have noticeably smaller parameter count ranging from 3.4M to 7.1M with the amount of parameters in the interpretable subject block reduced almost 10 times. It happens mainly due to the pruning of the subject block from original $K = 270$ branches to only $K = 6$. This allows for significantly better interpretability of the weights and makes the network less susceptible to overfitting which is extremely important considering the common issue in applications: a lack of large datasets with relevant brain activity data. On top of that the metrics stayed close to the baseline while models with 3 and 4 convolution blocks got stronger results than the baseline model despite being about 2 times smaller and more interpretable.

### 4.2 Ablation study

To check the impact of different layers in our architecture we tested a range of models with and without some of them. One can find the results in Table 1. The model from Défossez et al. (2023) uses 5 convolutional blocks after the subject block. We tried varying this parameter from 2 to 5 convolutional blocks. In our case the best models had 3 or 4 convolutional blocks depending on which accuracy metric one is more interested in. Then, we used our model but changed the 3D spatial attention into the original 2D one. The degradation of the metric values highlights the efficacy of the 3D spatial attention and aligns with the 2D vs 3D attention metrics on the baseline model available in the same table. Finally, we tried to add an additional temporal filter after the subject block. However, that did not lead to better results likely due to the transients at the edges of every epoch.

Table 1: Decoding results and ablations. Introducing the 3D spatial attention (SpAtt) not only improves accuracy, but also significantly reduces the amount of parameters of a model. The most notable difference happens in the interpretable subject block (ISB), reaching about a 10 fold decrease in parameters and permitting subsequent interpretation due to the use of the reduced count of latent sources $K = 6$, instead of $K = 270$ in the original work, see Figure 1.

| Model | K | Params | Params (ISB) | Top-1 | Top-10 |
|---|---|---|---|---|---|
| Défossez et al. (2023) | 270 | 9.65M | 2.6M | 42.28% | 72.64% |
| Défossez et al. (2023) + 3D SpAtt | 270 | 9.25M | 2.2M | **44.54**% | 72.12% |
| Ours (2 convs) | 6 | 3.4M | 0.27M | 41.54% | 71.01% |
| Ours (3 convs) | 6 | 4.6M | 0.27M | 43.17% | 72.7% |
| Ours (4 convs) | 6 | 5.8M | 0.27M | 42.83% | **73**% |
| Ours (5 convs) | 6 | 7.1M | 0.27M | 42.24% | 72.2% |
| Ours (5 convs + 2D SpAtt) | 6 | 7.2M | 0.43M | 41.88% | 71.98% |
| Ours (5 convs + temporal filter) | 6 | 7.1M | 0.27M | 38.77% | 69.63% |

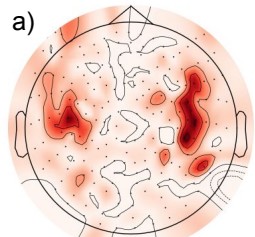 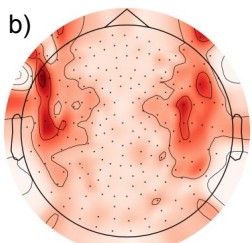

Figure 2: a) 2D and b) 3D spherical harmonics based total spatial attention

### 4.3 MODIFIED ATTENTION SCORES

Figure 2 shows the sum over all 270 receptive fields of the original 2D attention and the newly proposed 3D spherical harmonics based spatial attention layer. Physically justified spatial attention exhibits a smoother map highlighting the participation of not only the posterior temporal sources from the bilateral primary auditory cortices but also those in more anterior parts of the temporal lobe and Broca area of the frontal lobe. The above source localization judgments are very approximate as these attention maps can not be rigorously converted to the cortical distribution of sources due to the inherent non-linearity enforcing their elements to be strictly positive.

### 4.4 SPATIAL AND TEMPORAL PATTERNS

Figure 3 demonstrates the results of interpretation analysis of the network's subject block as shown in Figure 1. Each row of this plot corresponds to the $k$-th branch, $k = 1, .., 6$ of the interpretable subject's block and the rows are ordered based on the relevance as determined by the absolute gradient recipe proposed in (Petrosyan et al., 2021).

The left most column shows the topographies $\mathbf{g}_k$, $k = 1, ..., 6$ for the $K = 6$ interpretable branches. As described earlier and unlike Petrosyan et al. (2021) here compute the spatial filters $\mathbf{w}_k$ as the product of three matrices, see Figure 1 and Section 3.3, including the subject specific matrix which allows this architecture to aggregate information from the multi-subject dataset in a non-conflicting manner.

Note that in contrast to the spatial attention scores shown in Figure 2 the topographies $\mathbf{g}_k$ can be rigorously mapped to the source space as described in Section 2.2 and equation (5). The result of such mapping obtained using the MNE inverse solver (Gramfort et al., 2013) separately for each branch topography is shown in the last column of Figure 3. The middle column shows both normalized amplitudes response $|H_i(f)|$ of the temporal filters $\mathbf{h_i}$ and the power spectral density (PSD) profiles of the latent neuronal source corresponding to each of $K = 6$ branches.

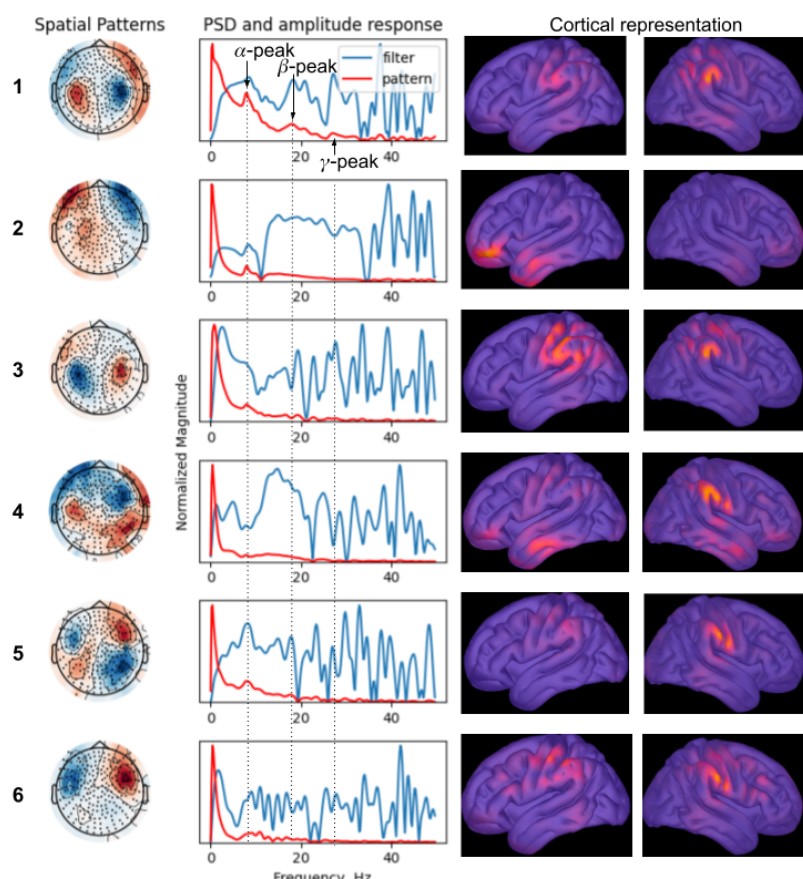

Figure 3: Spatial sensor-space (left), frequency domain profiles of the temporal patterns (middle) and cortical representation of the discovered latent sources (right) discovered by the network.

We can observe that the PSD of activity of all discovered sources has a significant amount of power in low frequencies, see red curves in Figure 3. However, analysis of the amplitude response of the filters (blue traces) shows that this low frequency was not found informative in the context of building self-supervised representation of neural activity induced by the natural speech stimulus in all but the 3-rd the 6-th branches. Bilateral sources in the temporal-parietal junction cortical area (components 1,5) and their activity in alpha and beta bands appeared the most important for the network. Components 3 corresponds to the spatially extended sources whose low frequency activity contributes to the classification task performed by the network. This observation appears plausible in the light of the hypotheses regarding the role of these slow rhythms in parsing the natural speech flow. Component 4 adds bilateral sources active in the extended upper beta-band in the middle temporal gyrus including the inferior temporal lobe and also the sources around the central sulcus known to host the tongue's sensory-motor representation whose beta band activity was selected by the training procedure. This component's importance for the classification task is consistent with the observation pushed forward by Bonilha et al. (2017) suggesting that the posterior lateral and inferior temporal cortices integrate auditory and conceptual processing crucial for auditory word comprehension. Component 2 most likely corresponds to a source of non-neuronal origin and reflects ocular-muscular activity, see https://www.fieldtriptoolbox.org/example/ica_ecg/, which is not surprising because of the known presence of involuntary eye-movements during narrative perception (Gehmacher et al., 2024; Braga et al., 2016). At the same time, since in this work we provide the averaged across subjects profiles this component's topography has other than ocular contributions. We therefore provided its cortical mapping and found sources in the frontal inferior temporal gyrus known to be a part of the speech comprehension system. Another interpretation of this component

can be rendered based on the recent report (Cope et al., 2023) who demonstrated that the temporal lobe perceptual predictions land in the inferior frontal cortex for reconciliation.

## 5 DISCUSSION

We have for the first time demonstrated the use of the explainable AI approach for mining the dynamic cortical representations of latent sources pivotal to the perceived speech decoding task from the non-invasive MEG data collected from a large cohort of subjects.

We augmented the network model presented in (Défossez et al., 2023) and endowed it with interpretable spatial and frequency domain selective layers which revealed cortical and dynamic representations of the latent sources discovered by the network. This was achieved by the interpretation of the network's weights in terms of the well accepted in the classical neurophysiological and neuroimaging communities notions of source topographies, activation time series and their second-order dynamical properties conveyed by the power spectral density.

We have also replaced the original 2D spatial attention layer with a more justified 3D spatial attention based on the spherical harmonics, a natural basis for representing magnetic fields in the vicinity of a volumetric conductor. Our main goal was to achieve the interpretability and after making sure that the introduced changes followed by significant (more than $40 \times$) pruning of the subject block did not significantly reduce the classification accuracy we focused on interpreting the sources that appeared pivotal for building self-supervised representations of the observed MEG activity.

In addition to the physilogically plausible patterns we also observed components of non-neuronal origin and related to eye movements. In some subjects (not reported) we found heart-beat related components which is unsurprising as naturalistic fictional stories cause emotionally driven variations in the heart rate (Wallentin et al., 2011; Beans, 2022).

## 6 LIMITATIONS

Our analyses and model development were based on a single MEG dataset but containing the data from 27 subjects, which may limit the generalizability of our findings.

The interpretation of our model relies on certain assumptions about linearity in the spatial and temporal filters; they may not fully capture the complexity of brain dynamics, potentially leading to oversimplified conclusions. Although our analysis results agree with the existing knowledge in the neuroscientific literature, the development of novel methods for analysis of subtle spatial-temporal dynamics of the discovered latent sources is warranted.

The architecture of our neural network, including the choice of spatial and temporal filters, is based on specific design choices that may have influenced the outcomes. We conducted a limited ablation study justifying our choices. The use of long temporal filters negatively affected the performance which is most likely due to the transients occurring at the edges of each new sample introducing. This needs to be ameliorated in the future designs. At the same time, as demonstrated in Petrosyan et al. (2021) despite performance degradation due to the added noise the interpretable network weights conveyed correct information regarding the dynamic properties of the underlying sources. Therefore in our interpretation we used network configuration containing the temporal filters.

Increasing the interpretability of our model by incorporating interpretable layers and filters has led to a trade-off with performance. While our modified model achieves comparable accuracy to the original, there is some slight reduction in decoding accuracy when 40 times fewer branches are used. Although not significant this highlights the balance that must be struck between model interpretability and performance.

We have not yet applied the individual head model based inverse modeling to the discovered subject specific topographies which is the work in progress and requires laborious steps of curating individual structural models obtained from the MRI of the participants.

## 7 ETHICS STATEMENT

The authors declare no competing interests. In the paper we utilize the publicly available dataset from Gwilliams et al. (2023). The study Gwilliams et al. (2023) was approved by the Institutional Review Board (IRB) ethics committee of New York University Abu Dhabi.

## 8 REPRODUCIBILITY STATEMENT

In this study we only utilize publicly available data. The details of our architecture and training are available in Section 3. The code will also be made available after the double blind review is completed.

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
