# OpenReview forum: "Perceived speech decoding and neurophysiological knowledge mining with explainable AI and non-invasive  brain activity recordings"
_ICLR.cc/2025/Conference — Submitted to ICLR 2025_

### Official Review · Reviewer_s2Qp · 2024-10-19

**Soundness:** 2
**Presentation:** 2
**Contribution:** 3
**Rating:** 3
**Confidence:** 3

**Summary:**

The manuscript proposed an interpretable CNN-variant model designed to decode a speech listening task, with the goal of uncovering spatial and temporal brain activity. To address the lack of temporal interpretability in the subject block of the baseline model, the authors introduce a temporal filter module that try to balance the performance and transparency. Additionally, the proposed 3D spatial attention module appears to enhance performance compared to the baseline model. Finally, the interpretable model is evaluated using a public MEG dataset from 27 subjects, and the authors discuss the resulting spatial and temporal patterns.

**Strengths:**

1. The proposed architecture enhances the model's trustworthiness and transparency for speech decoding, particularly in the temporal domain.
2. The authors used a natural way for MEG to capture information on a 3D spherical surface, called 3D spatial attention.

**Weaknesses:**

1. The abstract (line 18) mentioned that the model employed a self-supervised training strategy. However, I could not find any description or algorithm in the main body of the manuscript to support this claim, except for a brief mention in the discussion section (lines 469 and 507). This should be clarified or expanded upon in the methodology.
2. Although the 3D spatial attention contributes to improving only the top-1 accuracy of the baseline model, it does not seem to significantly reduce the model size, as indicated in the first two rows of Table 1. The size reduction appears to be mainly due to changes in the hyperparameters (K), rather than the spatial attention module itself. Additionally, it would be beneficial to include more ablation experiments on the 3D spatial attention module in 2-, 3-, and 4-layer models to verify its robustness.
3. Although the model demonstrates the ability to discover neuronal representations and qualitatively analyzes the discovered patterns, no quantitative metrics are used to evaluate whether the model's explanations align with existing meta-analyses. Given the inherent interpretability of the model and its goal of uncovering neuronal mechanisms, it would be valuable to confirm the effectiveness of these explanations with a more rigorous evaluation.
4. Only one dataset is used to verify the effectiveness and robustness of the proposed modules, which may not be sufficient. The baseline model was tested on four datasets (2 MEG + 2 EEG). Even though this study focuses on MEG data, at least one additional MEG dataset is available according to the baseline paper. Furthermore, the authors frequently mention the challenge of small MEG dataset sizes, but another available dataset seems to have a larger sample size. In particular, the authors themselves wrote (line 300):
“Note that given enough training data, the added temporal filters should not adversely affect the decoding accuracy as in the case when the frequency band specificity is not required the network can learn to be all-pass filters implementing the identity Hk (τ ) = δ(τ ) transform.”
Expanding the analysis to include a larger dataset would strengthen the claims about the model's generalizability.

**Questions:**

1. It seems that the authors split the dataset into training and testing datasets, but it is not clear how the hyperparameters (e.g., learning rate) were tuned using this split.

“This resulted in 2685 training and 999 testing segments which makes this study one of the few that utilize large datasets.” (line 341)

  More implementation details are needed, particularly regarding the optimizer, batch size, hidden layer size, and other training parameters. This level of detail is expected in deep learning research papers for reproducibility.

2. The choice of K=6, which is a crucial hyperparameter in the model, is not explained. Could you provide more detail on how and why this specific value was selected?
3. If I understand correctly, the negative impact of the temporal filter module on accuracy is reflected when comparing the sixth and eighth rows in Table 1, where a 3.47% accuracy drop is observed. Have you considered employing a post-hoc decision method for the temporal interpretability and potentially mitigate the performance decrease while maintaining interpretability?

---

### Official Review · Reviewer_68sB · 2024-10-26

**Soundness:** 1
**Presentation:** 2
**Contribution:** 3
**Rating:** 3
**Confidence:** 3

**Summary:**

This paper modifies the perceived speech decoding architecture for non-invasive neuroimaging data introduced by Défossez et al. (2023) to produce source topographies and power spectral density (PSD) profiles for the potential sources of latent neural representations of heard speech, following the framework of Petrosyan et al. (2021, 2022). The contribution is demonstrated through a comparison to the original architecture, a selection of ablations, and a comparison of the identified brain areas with current neuroscience literature.

**Strengths:**

The changes made to the architecture of Défossez et al. (2023) to leverage 3D spherical harmonics for spatial attention are strong. Additionally, highlighting the problem of lacking neuroscientific explainability with current non-invasive approaches is important.

**Weaknesses:**

It is unclear why the experimental decoding setup was changed from that of Défossez et al. (2023) despite relying on their model as a baseline. Additionally, the authors state that the testing segments are not aligned with word onset moments without offering a reason as to why this decision was made. This is concerning as Défossez et al. (2023) are intentional about doing this due to concerns about data leakage as the architecture relies on a contrastive loss between the representations of MEG data with the representations of the original auditory stimulus produced by a pretrained speech module. This concern is reinforced by the fact that the accuracy score reported for the paper's baseline implementation of Défossez et al. (2023)’s architecture is significantly higher than that of the original paper (72.64% vs 70.7%).

Additionally, results are not collected over multiple seeds so it is impossible to determine the effect of variance or evaluate whether the findings are statistically significant. The architecture used, as acknowledged by the original authors, is sensitive to hardware setup (i.e. number of GPUs used in training/testing) but these details are not given in the current paper. There is also no mention of hyperparameter selection or tuning, which draws into question the rigor of the effects found regarding model performance (alongside the concerns mentioned above).

**Questions:**

Suggestions to improve the paper:
- Address the issues with grammar and spelling throughout the paper
- In multiple places, things are stated with nothing more than a parenthetical that says “not reported” or “not described here”. It would be helpful if these results could be included in an appendix.
- Parts of the theoretical background could be moved to an appendix and simply referenced, leaving room for greater / missing details on implementation

---

### Official Review · Reviewer_V8FJ · 2024-10-28

**Soundness:** 1
**Presentation:** 1
**Contribution:** 1
**Rating:** 1
**Confidence:** 4

**Summary:**

This paper proposes an explainable artificial intelligence (XAI) approach for decoding perceived speech using non-invasive MEG data. It enhances an existing model (Défossez et al., 2023) by introducing a 3D spatial attention layer based on spherical harmonics and interpretable layers to map brain activity patterns and oscillations related to speech comprehension. The model achieves accuracy improvement to some extent.

**Strengths:**

1. This paper proposes 3D spatial attention and temporal filtering to improve previous method in decoding MEG signal to speech.

**Weaknesses:**

1. The writing is poor. Many sentences lack commas, which makes some paragraphs extremely hard to understand (e.g. line18-line23, line74-line76).
2. The contribution of methodology in this paper is rather limited. All the innovations are built upon the framework proposed by [1]. The authors simply (1) replace the original 2D spatial attention with 3D attention (2) add temporal filtering to the original model.
The work is incremental and not qualified for a top conference like ICLR. Moreover, the improvement brought by introduced methods is also quite limited from ablation study.
3. The motivation is confusing. The authors seek to incorporate explainable artificial intelligence in interpreting MEG-to-speech decoding. However, this paper lacks related analysis. The experiments only include the spatial attention and cortical representation analysis, which don't belong to XAI from my point of view. Almost all the papers related to computational neuroscience will conduct such experiments.

[1]. Decoding speech perception from non-invasive brain recordings. Nature Machine Intelligence

**Questions:**

Please refer to the weaknesses.

---

### Official Review · Reviewer_Zrwd · 2024-11-04

**Soundness:** 3
**Presentation:** 2
**Contribution:** 3
**Rating:** 5
**Confidence:** 3

**Summary:**

This paper is based on the prior work D ́efossez et al. (2023) and improved this architecture to identify latent neuronal sources linked to speech processing based on the MEG signal. The main contribution compared to the prior work is that the authors proposed a 3D spatial attention layer instead of the 2D version, applied a temporal filter to target specific frequency ranges, and reduced the number of latent sources. The numerical experiments show that the proposed method achieves comparable performance with the prior work with far less number of parameters.

**Strengths:**

This paper studies an important scientific question of identifying latent signal sources associated with speech processing from non-invasive MEG data. The authors made improvements upon prior work to incorporate 3D spatial filter using spherical harmonics and added a temporal filter. The experiments in Table 1 also show that reducing the number of latent sources to K=6 can greatly reduce the number of total parameters and potentially avoid overfitting. The results in Figures 2 and 3 demonstrate identified latent signal sources with interpretable neural functions.

**Weaknesses:**

1. The writing style involves long and packed sentences, which affect readability. The overall architecture is not discussed. I find it difficult to understand without reading the prior work D ́efossez et al. (2023).
2. The overall presentation leaves many important details unsaid. For example, the temporal filter h_k(\tau) is never properly defined, the evaluation criteria in Table 1 never explain the meaning of Top-1, Top-10.
3. It seems like the reduction in the number of parameters is mainly due to the specified number of latent sources decreasing from K=270 to K=6. However, the choice of K=6 seems arbitrary. The smoothed pattern shown in Figure 2 is a direct result of reducing K. Would reducing K lead to less precise localization of the latent signal sources? If Table 1 demonstrates the predictive performance, is there any sanity check on the accuracy of the identified signal locations? For example, if K =5 or 7, would the predictive performance stay the same? Would the identified sources be merged together with less K? If one reduces K to 6 for D ́efossez et al. (2023), would the prediction performance stay the same?

**Questions:**

In addition to the questions raised in weaknesses, I still have the following questions:
1. Equations (3) to (5) need to be clarified. How are \hat w_k estimated in equation (5)?
2. Is equation (3), it seems like the approximation for g_k ~ R w_k is based on w_k=(g_k^T R g_k)^{-1}R^{-1}g_k. If so, g_k would appear in the normalizing constant in equation (3), is the approximation g_k ~ R w_k still valid when ignoring the normalizing constant?

---

### Meta-Review · Area_Chair_2VPf · 2024-12-11

**Metareview:**

This submission contributes an explainable machine-learning architecture for prediction from MEG using a 3D spatial attention layer based on spherical harmonics. The main modification compared to a prior work is extending the 2D spatial attention layer to a 3D one. The reviewers found this contribution light from a methodological standpoint, and the manuscript lacking polish. Unfortunately, there was no discussion. With the light methodological contribution, it is not clear that this manuscript is suitable for ICLR.

**Additional Comments On Reviewer Discussion:**

No discussion, as the authors did not send a rebuttal.

---

### Decision · Program_Chairs · 2025-01-22

Reject